# Association Between Dietary Intake, Meal Patterns, and Malnutrition Risk Among Community-Dwelling Elderly in Northern Thailand: A Cross-Sectional Study

**DOI:** 10.3390/nu17223537

**Published:** 2025-11-12

**Authors:** Thanawit Sathittrakun, Narumit Bankhum, Phichayut Phinyo, Tanasit Wijitraphan, Yanee Choksomngam, Nalinee Yingchankul

**Affiliations:** 1Department of Family Medicine, Faculty of Medicine, Chiang Mai University, Chiang Mai 50200, Thailand; thanawit1995@gmail.com (T.S.); yanee.choksomngam@cmu.ac.th (Y.C.); 2Nutrition and Dietary Service Section, Faculty of Medicine, Chiang Mai University, Chiang Mai 50200, Thailand; kennutrition624@gmail.com; 3Department of Biomedical Informatics and Clinical Epidemiology (BioCE), Faculty of Medicine, Chiang Mai University, Chiang Mai 50200, Thailand; phichayutphinyo@gmail.com; 4Center for Clinical Epidemiology and Clinical Statistics, Faculty of Medicine, Chiang Mai University, Chiang Mai 50200, Thailand; 5San Pa Tong Hospital, Chiang Mai 50120, Thailand; tanasit.bird@gmail.com

**Keywords:** dietary intake, meal pattern, malnutrition, community-dwelling, older adults, vitamin E, fasting

## Abstract

**Background:** Malnutrition is prevalent among older adults in low- and middle-income countries (LMIC), but the roles of dietary intake and meal patterns are less understood in community settings. **Objective:** To investigate the associations between dietary intake, meal patterns, and malnutrition risk among community-dwelling older adults. **Methods:** We conducted a cross-sectional study including 200 adults aged ≥60 years in San Pa Tong District, Chiang Mai and used standardized questionnaires to collect sociodemographic, health, and dietary data. Nutritional status was assessed using the Mini Nutritional Assessment–Full Form (MNA-FF), with scores <24 indicated malnutrition risk. Meal patterns included overnight fasting duration, meal skipping, and eating frequency. Associations were examined using multivariable logistic regression with adjustment for confounding. **Results:** Overall, 23% of participants were at risk of malnutrition. Micronutrient inadequacy was widespread, and 92.5% exceeded the sodium upper limit. In multivariable models, higher vitamin E intake was independently associated with a lower likelihood of malnutrition (OR = 0.07, 95% CI: 0.01–0.45), whereas longer overnight fasting increased risk (OR = 1.94, 95% CI: 1.30–2.89) and having more eating episodes was protective (OR = 0.19, 95% CI: 0.06–0.57). **Conclusions:** Adequate vitamin E intake and regular meal patterns were associated with reduced malnutrition risk. Public health interventions should prioritize micronutrient adequacy, sodium reduction, and the promotion of regular eating patterns.

## 1. Introduction

Thailand became an aged society in 2022, with adults aged 60 years and older comprising over 20% of the population [1]. This demographic shift poses major public health challenges, particularly malnutrition, which contributes to frailty, functional decline, reduced quality of life, and increased healthcare costs [2]. National data show high rates of malnutrition and malnutrition risk among older adults, especially in Northern Thailand [3]. This condition arises from multiple causes (e.g., physiological, psychological, and socioeconomic), which collectively reduce dietary intake and promote frailty [2,3,4].

Dietary and meal patterns are increasingly recognized as determinants of nutritional health [5]. Diets rich in fruits, vegetables, whole grains, lean proteins, and healthy fats are associated with lower risks of non-communicable diseases, while processed foods and refined carbohydrates are linked to adverse metabolic outcomes [6,7,8]. Meal patterns, including eating frequency, timing, and regularity, influence energy and nutrient adequacy. Irregular intake has been associated with undernutrition, especially among older adults with reduced appetite or functional limitations [9,10]. Although time-restricted eating (TRE) and intermittent fasting (IF) have been extensively investigated in younger populations for their metabolic benefits [11], evidence in older adults regarding malnutrition and frailty remains limited [9].

Despite increasing research interest, few studies have explored how eating behaviors relate to malnutrition in community-dwelling older adults, as most have emphasized metabolic or weight-related outcomes. This is the first in Thailand to concurrently examine dietary intake and meal patterns in relation to malnutrition risk. It aims to clarify these associations among older adults in Northern Thailand to inform targeted interventions and public health strategies (Figure 1).

## 2. Materials and Methods

### 2.1. Study Design and Sample

This cross-sectional study was conducted from March to August 2024. Participants were recruited through convenience sampling from the service area of a public healthcare facility in San Pa Tong District, Chiang Mai Province. Trained healthcare staff used questionnaires provided by the research team to collect data. This process involved face-to-face interviews, body measurements, and physical performance tests.

Inclusion criteria were Thai adults aged ≥60 years who resided in San Pa Tong District. Recruitment was facilitated by local healthcare staff who served as research assistants, promoting the study at primary healthcare facilities and during community surveys across ten villages. A total of 200 participants were enrolled: 25% from healthcare facilities and 75% through community surveys. Exclusion criteria included inability to communicate in Thai, impaired consciousness that could affect data collection, or refusal to participate.

The study was conducted in accordance with the Declaration of Helsinki and approved by the Human Research Ethics Committee of San Pa Tong Hospital (protocol code SPT REC 029/2023).

### 2.2. Data Collection and Procedures

Data were collected through structured interviews, anthropometric measurements, and physical performance assessments. Sociodemographic variables included age, sex, marital status, education, income, occupation, religion, insurance, and living arrangements. Health-related data covered chronic diseases, medications, smoking, and alcohol use. Mental health was assessed using the Thai PHQ-9 (depression, cut-off score ≥10) [12] and the Mini-Cog (cognitive impairment defined as a memory score of 0 or 1–2 with abnormal clock drawing) [13].

Physical activity was measured using the Global Physical Activity Questionnaire (GPAQ), which captures activity in work, travel, and recreation [14]. Weekly duration was converted to MET-minutes (moderate activity = 4.0 METs, vigorous = 8.0 METs). Total energy expenditure (TEE) was estimated as REE × PAL, with resting energy expenditure (REE) calculated using the Mifflin–St Jeor equation [15,16].

Physical performance was evaluated using the Short Physical Performance Battery (SPPB), which assesses balance, walking speed, and chair stands (score range: 0–12); scores <10 indicate functional limitations and a higher risk of adverse outcomes [17,18]. Anthropometric measures (weight, height, BMI, mid-arm, waist, and calf circumference) were obtained using an electronic scale, a height measuring device, and flexible non-elastic tape. BMI was classified according to the Asia-Pacific classification criteria.

Nutritional status was assessed using the Mini Nutritional Assessment–Full Form (MNA-FF), a validated 18-item tool covering anthropometry, general health, dietary intake, and self-perception [19]. Scores ≥24 indicated well-nourished status, while scores <24 were classified as at risk of malnutrition.

Eating behavior was assessed in two domains: dietary intake and meal patterns. For all participants, dietary intake was evaluated using both a 24 h dietary recall (24HR) and a semi-quantitative Food Frequency Questionnaire (FFQ). The 24HR recorded all foods, snacks, and beverages consumed on the previous day, while the FFQ captured habitual consumption frequencies of various food items. To enhance accuracy, trained interviewers used standardized probing techniques and picture aids to help participants recall foods and estimate portion sizes using common household measures (e.g., cups, teaspoons, ladles, glasses).

Eating behaviors were classified as follows: (1) overnight fasting, defined as the interval from the last meal before sleep to the first meal after waking (including main meals and snacks); (2) number of eating episodes, defined as instances of consuming ≥50 kcal at least one hour apart; and (3) meal skipping, defined as consuming <200 kcal or eating outside designated timeframes (6:00–9:00 a.m. for breakfast, 11:00 a.m.–2:00 p.m. for lunch, 5:00–8:00 p.m. for dinner). A registered dietitian used INMUCAL-Nutrients version 4.0 (Institute of Nutrition, Mahidol University, Thailand) to analyze daily caloric, macronutrient, and micronutrient intake from the 24HR and FFQ data.

### 2.3. Sample Size Calculation

The sample size was calculated using the infinite population proportion formula, based on an estimated malnutrition prevalence of 11.25% among older adults in Northern Thailand [3]. With a 95% confidence level and a 5% margin of error, the required sample size was 153. To account for potential non-response or incomplete data, the final target was increased to 184 participants.

### 2.4. Statistical Analysis

All analyses were conducted using Stata/MP version 17.0 (StataCorp, College Station, TX, USA). Descriptive statistics were used to summarize sociodemographic, health-related, and eating behavior variables. Group differences between participants who were well-nourished and those at risk of malnutrition were assessed using Fisher’s exact or chi-square tests for categorical variables, and independent *t*-tests or Mann–Whitney U tests for continuous variables, depending on data distribution. Statistical significance was set at *p* < 0.05.

To examine the associations between dietary intake, meal patterns, and malnutrition risk, multivariable logistic regression analyses were conducted. Macronutrient and micronutrient variables were standardized by dividing each by its respective standard deviation to allow for effect size comparison [20]. Four models were constructed: Model 0 was the crude (unadjusted) model; Model 1 was adjusted for age, sex, marital status, and monthly income; Model 2 was additionally adjusted for cognitive impairment and functional status (non-collinear variables significantly associated with malnutrition risk in univariate analyses); and Model 3 was further adjusted for total energy intake. For macronutrient analyses, energy intake from the macronutrient of interest was excluded from the adjustment to avoid collinearity. Results were reported as odds ratios (ORs) with 95% confidence intervals (CIs).

## 3. Results

Among the 200 participants, the prevalence of malnutrition risk (MNA < 24) was 23%. Participants at risk of malnutrition were older than well-nourished individuals (73.2 vs. 67.1 years). There were significant sociodemographic differences between groups, including higher proportions of unemployment (71.7% vs. 49.4%) and low income (87.0% vs. 49.4%). Based on PHQ-9 scores (range 0–5), no participants met the criteria for depression. Alcohol consumption and smoking were infrequent, reported by 13.5% and 3.5% of participants, respectively. There were also significant health-related differences, including insufficient physical activity (65.2% vs. 32.5%), cognitive impairment (37.0% vs. 4.6%), and poor physical performance (65.2% vs. 19.5%) (Table 1).

Eating behaviors are summarized in Table 2. The mean daily energy intake was 1609.8 ± 505.4 kcal, with dinner contributing the largest proportion. Carbohydrate intake was significantly higher in the well-nourished group, particularly at breakfast and dinner. Most participants consumed three meals per day, with a median overnight fasting duration of 13.5 h. The majority (83%) did not skip main meals.

Table 3 compares nutrient intake with the Thai Dietary Reference Intake (DRI) 2020. Adequacy was defined using recommended dietary allowance (RDA) or adequate intake (AI) cutoffs, depending on the nutrient. Only 38.5% of participants achieved an adequate energy intake (≥TEE), which varies based on individual physical activity levels and basal energy expenditure. Nearly all (99.5%) met the protein RDA, while 39.0% and 84.0% met the RDAs for carbohydrates and fat, respectively. Both groups demonstrated widespread inadequacies in vitamin A, vitamin C, vitamin E, vitamin B6, vitamin B12, calcium, magnesium, zinc, selenium, copper, potassium, and dietary fiber, with more than 50% of participants falling below recommended levels. In contrast, 92.5% exceeded the sodium upper intake level. The full table of nutrient intake and adequacy is provided in Appendix A.

Associations between dietary factors and malnutrition risk are presented in Table 4. Higher energy intake was inversely associated with malnutrition risk in the crude model (OR = 0.66, 95% CI: 0.47–0.93), but this association was not significant after adjustment. Higher intake of carbohydrates (OR = 0.56, 95% CI: 0.37–0.84), vitamin B6 (OR = 0.57, 95% CI: 0.32–1.00), and magnesium (OR = 0.61, 95% CI: 0.38–0.97) remained statistically significant after adjustment for age, sex, marital status, and low income (Model 1); but lost significance after further adjustment for cognitive impairment and poor physical performance (Model 2), although the effect sizes remained similar. Protein and fat intake showed no associations in any model. Among micronutrients, only vitamin E maintained a significant inverse association in the fully adjusted model (OR = 0.07, 95% CI: 0.01–0.45).

Regarding meal patterns, longer overnight fasting was consistently associated with a higher risk of malnutrition across all models (OR = 1.94, 95% CI: 1.30–2.90), while a greater number of eating episodes was inversely associated in all models (OR = 0.19, 95% CI: 0.06–0.58). Meal skipping was significantly associated with malnutrition risk in Models 0 and 1 (OR = 2.52, 95% CI: 1.01–6.27), but the association lost statistical significance in Models 2 and 3.

## 4. Discussion

The prevalence of malnutrition risk in this study was lower than previously reported in other Thai cohorts (36.0–54.8%) [3,21], which is likely due to the relatively healthier and younger sample. Factors such as lower alcohol and tobacco use, absence of depression, and better physical performance may have contributed to this reduced risk [22,23].

Adequate energy and macronutrient intake are well-established protective factors against malnutrition [24,25]. In older adults, appetite loss, reduced lean mass, low activity, and disease-related malabsorption increase vulnerability [25]. In our sample, energy and macronutrient intake exceeded those reported in the 5th Thai National Food and Nutrition Survey, 2003 [26], which may partly explain the lower prevalence observed. Beyond their caloric contribution, protein plays a critical role in maintaining muscle mass, supporting tissue repair, and preventing frailty and sarcopenia [27,28], while carbohydrates serve as the primary and most readily accessible source of energy [25]. Excessive carbohydrate restriction can reduce intake of vitamin B, minerals, and antioxidants, thereby increasing the risk of malnutrition [29]. Fat remains essential for the absorption of fat-soluble vitamins, hormone synthesis, and membrane integrity [30]. However, protein and fat showed no association with malnutrition in our models, likely because intake was uniformly high, with 99.5% and 84% of participants meeting the RDA, respectively. This limited variability may have reduced the statistical power to detect meaningful differences. From a practical standpoint, older adults should maintain sufficient but balanced energy and macronutrient intake to prevent both undernutrition and overnutrition.

Micronutrient deficiencies in later life are associated with cognitive decline, reduced physical activity, impaired immunity, and increased burdens of chronic disease (e.g., hypertension, diabetes, osteoporosis) [31,32]. Although multiple factors contribute, insufficient food intake remains a primary driver [33]. Global evidence highlights common inadequacies in vitamin D, B1, B2, B9, B12, calcium, magnesium, selenium, and iron among older adults [31,33]. In Thailand, older adults frequently consume insufficient amounts of vitamins A, C, E, B1, B2, B3, calcium, phosphorus, and fiber [26,32,34,35]. Consistent with these data, more than half of our participants had inadequate intake of vitamins A, C, E, B6, B12, as well as calcium, magnesium, iron, and fiber. A plausible contributor is the low consumption of milk and dairy products among older Thai adults, which may lead to inadequate intake of several essential vitamins and minerals, particularly calcium, phosphorus, magnesium, and vitamin B12 [26].

Higher vitamin E intake was independently associated with a reduced risk of malnutrition. Vitamin E is a lipid-soluble antioxidant, primarily α-tocopherol, that protects polyunsaturated fatty acids within cell membranes and lipoproteins from lipid peroxidation [36,37]. Deficiency increases oxidative stress and disrupts redox balance, leading to structural damage of membranes and mitochondria, activation of pro-inflammatory pathways (e.g., NF-κB, AP-1), and impairment of immune function [37,38]. These cellular disturbances can manifest clinically as neuropathy, ataxia, and anemia [36,37]. In muscle tissue, excess oxidative stress promotes proteolysis through activation of the ubiquitin–proteasome system and mitochondrial dysfunction, thereby contributing to sarcopenia and frailty, which are closely intertwined with malnutrition [39]. Absorption and transport of vitamin E require adequate dietary fat and functional α-tocopherol transfer protein; consequently, sustained low-fat intake, protein–energy deficiency, or fat malabsorption can compromise bioavailability [37]. Dietary sources include vegetable oils, nuts, whole grains, and green leafy vegetables [40,41], but older adults often consume limited amounts of these foods due to chewing or swallowing difficulties, increasing their risk of inadequacy [42]. Collectively, these mechanisms provide biological plausibility for the observed association between low vitamin E intake and malnutrition risk in this study.

Interestingly, vitamin E intake differed significantly between the well-nourished and malnutrition risk groups, despite comparable total fat intake. This finding likely reflects differences in the sources and quality of dietary fat. Older adults with better nutritional status may consume more plant-based oils, nuts, and seeds, which are rich sources of vitamin E, whereas those at higher risk may rely more on animal fats or fried foods with lower vitamin E content. Thus, vitamin E intake may serve as a marker of overall diet quality, as individuals who consume more vitamin E generally have higher intake of vegetables, whole grains, and other nutrient-dense foods.

Excess sodium intake was highly prevalent (>90% of participants), reflecting global concerns about the dual burden of malnutrition and noncommunicable diseases. In Thailand, taste alterations and a preference for strongly seasoned foods contribute to high sodium intake; for example, two-thirds of older adults add condiments to noodle dishes [26]. High sodium intake is associated with increased risks of cardiovascular disease and mortality [25]. In this study, sodium intake was inversely associated with malnutrition risk in the crude model, likely reflecting greater overall food consumption among well-nourished individuals. However, this association was no longer significant after statistical adjustment.

Intermittent fasting has been promoted for metabolic health benefits in younger populations [11]. In contrast, our findings demonstrate that prolonged overnight fasting in older adults is consistently associated with increased malnutrition risk, consistent with previous evidence [43]. Prolonged fasting may accelerate protein catabolism, exacerbate frailty, and impair recovery from illness. Based on these results, overnight fasting durations of 13.5 h or longer should be avoided. National survey data (NHES IV) indicate that 20% of Thai older adults skip meals, most often lunch [44], while in our study, breakfast and dinner were skipped more frequently, with dinner contributing the largest proportion of daily energy intake. Skipping these meals prolongs the fasting interval and reduces eating frequency. Consistent with Nordic recommendations, overnight fasting in older adults should not exceed 11 h, and at least four eating episodes per day are recommended [5].

From a public health perspective, these findings underscore the need to expand screening coverage for older adults to facilitate early detection of malnutrition, which remains insufficient in many areas. Strategies should also emphasize the promotion of antioxidant-rich foods, regular meal patterns, and reduced sodium intake. These priorities align with Thailand’s forthcoming sodium tax initiative aimed at lowering salt consumption. Incorporating such measures into primary healthcare and community programs could enhance preventive nutrition and support healthy aging in Thailand and other low- and middle-income country (LMIC) settings.

This study has several limitations. First, the cross-sectional design precludes causal inference, and the observed associations should therefore be interpreted with caution. Second, dietary assessment was based on a single 24-h recall supplemented by an FFQ, which may not adequately capture day-to-day or seasonal variability. Although recall bias is a common issue among older adults, this limitation was mitigated by the use of trained interviewers, standardized probing techniques, and picture aids to enhance reporting accuracy. Third, convenience sampling from community centers may limit the generalizability of findings to the broader older adult population in Thailand. Finally, the relatively low prevalence of malnutrition risk (23%) may have reduced the events-per-variable ratio in fully adjusted models. The sensitivity of results to covariate adjustment (with loss of statistical significance in Models 2 and 3) suggests potential overfitting and residual confounding. Future studies should employ longitudinal designs with multi-day and seasonal dietary assessments, complemented by pre-registered sensitivity analyses, to strengthen causal inference and external validity.

Despite these limitations, the study has notable strengths. It employed validated nutritional and functional assessment tools and accounted for multiple sociodemographic and health-related confounders, thereby enhancing the robustness of the findings. Importantly, this is the first study in Thailand to concurrently examine dietary intake and meal patterns in relation to malnutrition risk among community-dwelling older adults. The results provide context-specific evidence that can inform the design of future nutritional interventions and public health policies.

## 5. Conclusions

This study demonstrated that higher vitamin E intake was independently associated with a reduced risk of malnutrition, while prolonged overnight fasting and fewer daily eating episodes increased risk among community-dwelling older adults in Northern Thailand.

These findings underscore the importance of routine nutritional screening and community-based interventions that promote vitamin E-rich foods (such as nuts, seeds, and vegetable oils), encourage regular meal consumption, avoid prolonged fasting, and reduce sodium intake to support healthy aging. Given Thailand’s rapidly aging population and similar demographic transitions in other low- and middle-income countries, these results provide timely evidence to inform nutritional policy and highlight the need for longitudinal and interventional studies to confirm causal relationships.

## Figures and Tables

**Figure 1 nutrients-17-03537-f001:**
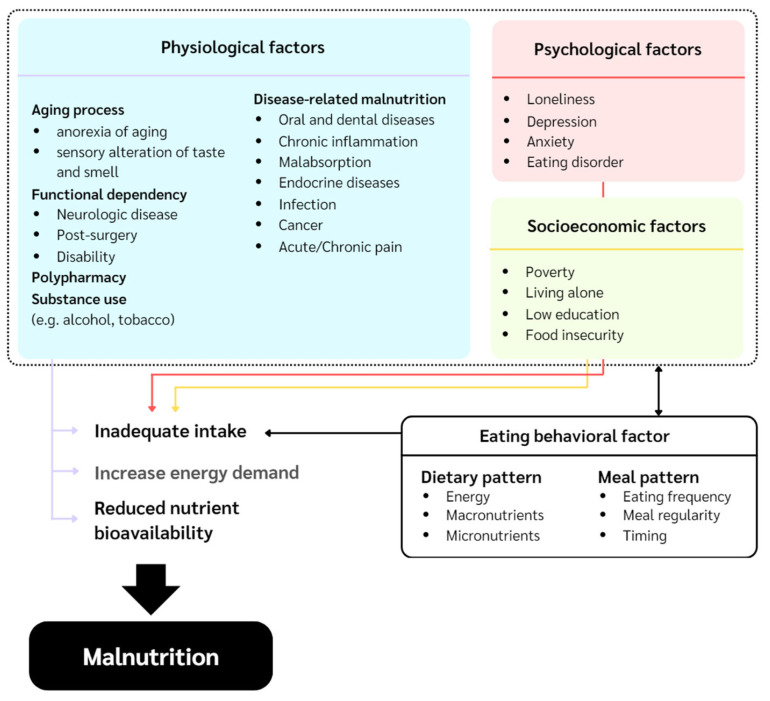
Conceptual framework for determinants of malnutrition in older adults.

**Table 1 nutrients-17-03537-t001:** Descriptive characteristics of the participants according to nutritional status.

Characteristics	Total (N = 200)	Malnutrition Risk (N = 46)	Well-Nourished (N = 154)	*p*-Value
Age (years), mean ± SD	68.5 ± 6.5	73.2 ± 8.5	67.1 ± 5.0	**<0.001** ^a^
Sex: Female	122 (61.0)	26 (56.5)	96 (62.3)	0.478 ^b^
Single/Separated/Divorced/Widowed	72 (36.0)	22 (37.8)	50 (32.5)	0.057 ^b^
Education level ≤ Elementary school	159 (79.5)	41 (89.1)	118 (76.6)	0.065 ^b^
Unemployment	109 (54.5)	33 (71.7)	76 (49.4)	**0.007** ^b^
Income ≤ 5000 bath/month (153 USD/month)	116 (58.0)	40 (87.0)	76 (49.4)	**<0.001** ^b^
Multimorbidity	77 (38.5)	18 (39.1)	59 (38.3)	0.920 ^b^
Polypharmacy (≥5 types of medication used)	13 (6.5)	5 (10.9)	8 (5.2)	0.181 ^c^
Active smokers	7 (3.5)	3 (6.5)	4 (2.6)	0.200 ^c^
Alcohol drinkers	27 (13.5)	5 (13.0)	18 (13.6)	0.918 ^b^
Cognitive impairment	24 (12.0)	17 (37.0)	7 (4.6)	**<0.001** ^b^
Physical activity, MET-minutes, median (IQR)	790 (350, 1400)	360 (0, 960)	800 (440, 1600)	**<0.001** ^d^
Inadequate physical activity (<600 MET-minute)	80 (40.0)	30 (65.2)	50 (32.5)	**<0.001** ^b^
TEE (kcal), median (IQR)	1744.7 (1429.3, 2068.5)	1379.8 (1171.1, 1750.6)	1840.5 (1515.9, 2135.6)	**<0.001** ^d^
SPPB score, median (IQR)	11 (8, 12)	7 (3, 10)	11 (10, 12)	**<0.001** ^d^
10–12 (well physical performance)	140 (70.0)	16 (34.8)	124 (80.5)	**<0.001** ^b^
0–9 (poor physical performance)	60 (30.0)	30 (65.2)	30 (19.5)	

SD, standard deviation; IQR, interquartile range; TEE, total energy expenditure; SPPB, short physical performance battery. Significant results are shown in bold. ^a^ Independent *t*-test, ^b^ Chi-square test, ^c^ Fisher’s exact test, ^d^ Mann–Whitney U test.

**Table 2 nutrients-17-03537-t002:** Association between dietary intake and meal patterns with nutritional status.

Characteristics	Total (N = 200)	Malnutrition Risk (N = 46)	Well-Nourished (N = 154)	*p*-Value
Dietary pattern				
Daily calorie intake (kcal), mean ± SD	1609.77 ± 505.43	1453.27 ± 546.98	1656.51 ± 484.47	**0.016** ^a^
Breakfast calorie intake	487.67 ± 309.63	486.14 ± 363.43	488.13 ± 292.99	0.970 ^a^
Lunch calorie intake	513.62 ± 234.04	465.68 ± 231.61	527.93 ± 233.60	0.114 ^a^
Dinner calorie intake	579.20 ± 316.461	491.15 ± 292.43	605.50 ± 319.50	**0.031** ^a^
Protein daily intake (gm), median (IQR)	91.03 (63.61, 124.31)	100.84 (51.37, 127.35)	87.81 (64.12, 123.80)	0.972 ^d^
Breakfast	22.69 (10.12, 46.16)	21.46 (7.55, 53.16)	23.03 (10.48, 42.66)	0.712 ^d^
Lunch	25.26 (13.66, 42.84)	22.58 (11.09, 43.03)	25.91 (14.30, 42.66)	0.544 ^d^
Dinner	31.63 (17.31, 53.15)	31.89 (12.80, 46.76)	31.63 (18.48, 53.64)	0.387 ^d^
Carbohydrate daily intake (gm), median (IQR)	148.13 (109.56, 198.36)	116.79 (85.41, 165.98)	154.30 (122.43, 202.73)	**0.002** ^d^
Breakfast	46.94 (23.89, 65.01)	35.93 (15.03, 63.77)	49.28 (27.21, 65.40)	**0.030** ^d^
Lunch	50.97 (30.02, 76.32)	45.08 (23.41, 68.45)	52.36 (35.76, 76.65)	0.075 ^d^
Dinner	47.08 (32.06, 66.87)	39.01 (24.06, 55.04)	50.97 (35.16, 69.65)	**0.006** ^d^
Fat daily intake (gm), median (IQR)	54.44 (36.59, 83.52)	51.21 (31.26, 80.99)	54.58 (38.78, 84.36)	0.344 ^d^
Breakfast	10.75 (3.00, 30.46)	10.58 (3.05, 30.27)	10.85 (2.95, 30.55)	0.977 ^d^
Lunch	13.17 (5.88, 28.37)	11.86 (3.81, 27)	13.17 (6.74, 29.77)	0.209 ^d^
Dinner	19.17 (7.44, 34.50)	17.15 (2.77, 31.13)	19.31 (8.42, 34.63)	0.164 ^d^
Meal pattern				
Overnight fasting (h), median (range)	13.5 (10.5–23)	14 (11.5–23)	13.5 (10.5–17.5)	**<0.001** ^d^
Eating episodes * (times), median (range)	3 (1–6)	3 (1–4)	3 (2–6)	**<0.001** ^d^
Meal skipper **, N (%)	34 (17.00)	14 (30.43)	20 (12.99)	**0.006** ^b^
Breakfast skipper	25 (12.50)	9 (19.57)	16 (10.39)	0.099 ^b^
Lunch skipper	4 (2.00)	2 (4.35)	2 (1.30)	0.2270 ^c^
Dinner skipper	6 (3.00)	4 (8.70)	2 (1.30)	**0.026** ^c^

* An ‘eating episode’ is considered if a minimum of 50 calories are consumed, and the meal interval is more than 1 h (count both meal and snack) ** Meal skipper is classified if consuming less than 200 calories within designated timeframes: 6:00 AM–9:00 AM (breakfast), 11:00 AM–2:00 PM (lunch), or 5:00 PM–8:00 PM (dinner). Significant results are shown in bold. ^a^ Independent *t*-test, ^b^ Chi-square test, ^c^ Fisher’s exact test, ^d^ Mann–Whitney U test.

**Table 3 nutrients-17-03537-t003:** Prevalence of inadequate nutrient intake according to the Thai Dietary Reference Intake 2020.

Nutrients	Total (N = 200)	Malnutrition Risk (N = 46)	Well-Nourished (N = 154)	*p*-Value
TEI (kcal/d), mean ± SD	1609.8 ± 505.4	1453.3 ± 547.0	1656.5 ± 484.5	**0.016** ^a^
TEI < TEE	123 (61.5)	23 (50)	100 (64.9)	0.068 ^b^
Macronutrients				
Protein (g/d), mean ± SD	99.3 ± 46.9	98.2 ± 50.2	99.6 ± 46.0	0.855 ^a^
%TEI, median (range)	23.34 (19.76, 28.70)	24.79 (20.35, 29.15)	22.97 (19.75, 28.44)	0.158 ^d^
<10% of TEI	1 (0.5)	0 (0)	1 (0.7)	1.000 ^c^
Carbohydrates (g/d), mean ± SD	159.5 ± 72.8	132.4 ± 67.2	167.6 ± 72.6	**0.004** ^a^
%TEI, median (range)	40.22 (29.22, 52.99)	35.72 (26.63, 52.84)	40.62 (29.80, 53.13)	0.170 ^d^
<45% of TEI	122 (61.0)	32 (69.6)	90 (58.4)	0.175 ^b^
Fat (g/d), mean ± SD	62.8 ± 36.7	59.0 ± 39.0	63.9 ± 36.1	0.427 ^a^
%TEI, median (range)	33.15 (23.41, 44.47)	33.53 (24.09, 46.82)	33.06 (23.06, 42.93)	0.778 ^d^
<20% of TEI	32 (16.0)	7 (15.2)	25 (16.2)	0.869 ^b^
Micronutrients				
Vitamin A (mcg/d), median (range)	218.19 (78.21, 459.75)	218.83 (75.63, 423.71)	218.19 (83.05, 475.24)	0.678 ^d^
<RDA	170 (85.0)	42 (91.3)	128 (83.1)	0.172 ^b^
Vitamin C (mg/d), median (range)	42.14 (16.26, 92.24)	36.20 (18.91, 88.74)	46.83 (15.68, 102.36)	0.477 ^d^
<RDA	147 (73.5)	35 (76.1)	112 (72.7)	0.651 ^b^
Vitamin E (mg/d), median (range)	0.91 (0.09, 2.44)	0.12 (0, 1.20)	1.05 (0.16, 2.62)	**<0.001** ^d^
<RDA	197 (98.5)	46 (100)	151 (98.1)	1.000 ^c^
Calcium (mg/d), median (range)	313.34 (179.48, 582.14)	292.33 (136.50, 603.50)	318.53 (213.79, 570.76)	0.357 ^d^
<RDA	185 (92.5)	44 (95.7)	141 (91.6)	0.528 ^c^
Magnesium (mg/d), median (range)	19.99 (5.76, 58.57)	10.68 (0.53, 39.51)	24.55 (7.56, 63.79)	**0.008** ^d^
<RDA	199 (99.5)	46 (100)	153 (99.4)	1.000 ^c^
Iron (mg/d), median (range)	11.29 (8.00, 14.33)	9.67 (6.98, 15.14)	11.69 (8.24, 15.52)	0.061 ^d^
<RDA	90 (45.0)	27 (58.7)	63 (40.9)	**0.033** ^b^
Zinc (mg/d), median (range)	5.99 (4.08, 8.18)	4.57 (3.28, 6.95)	6.39 (4.31, 8.40)	**0.010** ^d^
<RDA	168 (84.0)	41 (89.1)	127 (82.5)	0.279 ^b^
Sodium (mg/d), median (range)	3744.61 (2524.8, 5898.36)	3193.48 (2004.67, 5102.23)	3979.27 (2738.48, 5933.14)	**0.039** ^d^
>UL	185 (92.5)	43 (93.5)	142 (92.2)	0.774 ^b^
Dietary fiber (g/d), median (range)	9.05 (4.63, 13.29)	8.81(3.09, 11.45)	9.39 (5.50, 13.69)	**0.034** ^d^
<RDA	188 (94.0)	45 (97.8)	143 (92.9)	0.303 ^c^

RDA, recommended dietary allowances; TEI, total energy intake; TEE, total energy expenditure. Significant results are shown in bold. ^a^ Independent *t*-test, ^b^ Chi-square test, ^c^ Fisher’s exact test, ^d^ Mann–Whitney U test.

**Table 4 nutrients-17-03537-t004:** Association of dietary intake (divided by the standard deviation) and meal pattern with malnutrition risk.

Variables	Model 0		Model 1		Model 2		Model 3	
	OR (95% CI)	*p*-Value	OR (95% CI)	*p*-Value	OR (95% CI)	*p*-Value	OR (95% CI)	*p*-Value
TEI (SD = 505.43 kcal/d)	0.66 (0.47–0.93)	**0.018**	0.71 (0.48–1.06)	0.096	0.85 (0.56–1.29)	0.443	-	-
Macronutrients								
Protein (SD = 46.88 g/d)	0.96 (0.70–1.35)	0.854	1.00 (0.68–1.46)	0.989	1.14 (0.75–1.71)	0.540	1.35 (0.86–2.10)	0.189
Carbohydrate (SD = 72.78 g/d)	0.56 (0.37–0.84)	**0.005**	0.63 (0.40–0.98)	**0.038**	0.77 (0.48–1.24)	0.292	0.76 (0.47–1.23)	0.268
Fat (SD = 36.72 g/d)	0.87 (0.62–1.22)	0.425	0.91 (0.62–1.35)	0.641	0.93 (0.62–1.39)	0.719	0.95 (0.63–1.44)	0.818
Micronutrients								
Vitamin E (SD = 5.82 mg/d)	0.24 (0.06–0.96)	**0.044**	0.08 (0.02–0.45)	**0.004**	0.07 (0.01–0.44)	**0.004**	0.07 (0.01–0.45)	**0.005**
Vitamin B6 (SD = 0.42 mg/d)	0.64 (0.39–1.07)	0.086	0.57 (0.32–1.00)	**0.049**	0.58 (0.33–1.03)	0.065	0.59 (0.33–1.06)	0.075
Magnesium (SD = 53.02 mg/d)	0.74 (0.48–1.13)	0.163	0.61 (0.38–0.97)	**0.039**	0.73 (0.46–1.18)	0.200	0.75 (0.46–1.21)	0.235
Iron (SD = 8.84 mg/d)	0.58 (0.35–0.97)	**0.038**	0.63 (0.37–1.08)	0.091	0.69 (0.39–1.21)	0.195	0.71 (0.39–1.29)	0.258
Sodium (SD = 3599.05 mg/d)	0.61 (0.38–0.98)	**0.042**	0.70 (0.42–1.17)	0.172	0.71 (0.42–1.21)	0.207	0.74 (0.43–1.26)	0.265
Meal pattern								
Overnight fasting duration	1.86 (1.35–2.54)	**<0.001**	1.97 (1.36–2.86)	**<0.001**	1.95 (1.30–2.91)	**0.001**	1.94 (1.30–2.89)	**0.001**
Meal skipper	2.99 (1.41–6.35)	**0.004**	2.52 (1.01–6.27)	**0.047**	2.03 (0.78–5.29)	0.149	1.92 (0.69–5.33)	0.208
Eating episodes	0.20 (0.08–4.80)	**<0.001**	0.15 (0.05–0.45)	**0.001**	0.19 (0.06–0.56)	**0.003**	0.19 (0.06–0.57)	**0.003**

TEI, total energy intake. Significant results are shown in bold. Model 0: Crude. Model 1: adjusted for age, sex, marital status, low income. Model 2: adjusted for variables in model 2 plus cognitive impairment, poor physical performance. Model 3: adjusted for variables in model 3 plus total energy intake.

## Data Availability

The original contributions presented in this study are included in the article/Appendix A. Further inquiries can be directed to the corresponding author.

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
