# Peer review of "Association Between Dietary Intake, Meal Patterns, and Malnutrition Risk Among Community-Dwelling Elderly in Northern Thailand: A Cross-Sectional Study"

_nutrients, 2025, doi:10.3390/nu17223537_

Round 1

Reviewer 1 Report

Comments and Suggestions for Authors

General Comments
The manuscript presents a relevant and timely topic addressing dietary intake, meal patterns, and malnutrition risk among community-dwelling older adults in Northern Thailand. The study is well structured, methodologically sound, and supported by clear statistical analyses. The sample size is appropriate, and the statistical approach is generally robust. However, several sections would benefit from clearer writing, removal of redundancies, better alignment of results and conclusions, and more consistent formatting throughout the text. These changes would enhance the readability and scientific impact of the paper.

Specific Comments

Title and Abstract
- Lines 2–4: Simplify the title to reduce redundancy. Remove or reorder terms such as 'high' or 'risk' to make the sentence clearer.
- Lines 15–34: Include effect sizes and confidence intervals in the abstract alongside OR values.
- Lines 29–31: Rephrase 'may reduce malnutrition risk' to 'was associated with reduced malnutrition risk' for greater scientific precision.

Introduction
- Lines 37–44: Merge short and repetitive sentences regarding aging and malnutrition to create a more direct argument.
- Lines 58–63: Remove '(figure 1)' from the middle of the sentence and reposition it more clearly at the end of the paragraph.
- Line 62: Correct spacing ('northern Thailand ,') to 'Northern Thailand,'.

Materials and Methods
- Lines 68–74: Simplify the description of geographic location by removing unnecessary travel details.
- Lines 98–100: Correct repetition ('obtained using were collected using') to 'obtained using'.
- Lines 117–122: Streamline sample size calculation text to remove repeated terms.
- Lines 123–130: Add justification for statistical significance level or expected effect size.
- Line 129: Replace 'P-value < 0.05' with 'p < 0.05' for consistency.
- Line 133: Rephrase 'to allow for comparison of effect sizes' to 'to standardize variables for effect size comparison'.

Results
- Lines 143–146: Standardize terminology, replacing 'well nutrition' with 'well-nourished'.
- Lines 154–155: Simplify and break long sentences regarding eating episodes and fasting for better clarity.
- Lines 158–164: Correct grammar: replace '. which' with ', which'.
- Table 2 (Lines 183–189): Standardize p-value formatting and align columns properly.

Discussion
- Lines 206–211: Rewrite to remove unnecessary enumerations and improve flow.
- Lines 240–245: Rephrase to be more concise ('Vitamin E is a lipid-soluble antioxidant, primarily α-tocopherol, that protects...').
- Lines 258–265: Improve cohesion and separate the long sentence about sodium intake and adjustment effects into two.

Conclusion
- Lines 295–304: Rephrase to make practical recommendations more explicit (e.g., 'promoting vitamin E-rich foods, avoiding prolonged fasting, and reducing sodium intake').

References and Formatting
- References 6 and 45: Remove duplicates references
- References 8 and 364–366: Correct inconsistent capitalization.
- Ensure uniform formatting of journal titles, volumes, and page numbers.

Author Response

Comment 1:

Title and Abstract

  • Lines 2–4: Simplify the title to reduce redundancy. Remove or reorder terms such as 'high' or 'risk' to make the sentence clearer.
  • Lines 15–34: Include effect sizes and confidence intervals in the abstract alongside OR values.
  • Lines 29–31: Rephrase 'may reduce malnutrition risk' to 'was associated with reduced malnutrition risk' for greater scientific precision.

Response: Thank you for your suggestion.

  • We have simplified the title by changing “high malnutrition risk” to “malnutrition risk” and have used this term consistently throughout the manuscript.
  • The abstract has been revised to be more concise, and effect sizes (OR, 95% CI) have been added (Page 1, Lines 29–31).
  • We have also revised the sentence in the abstract to read: “Adequate vitamin E intake and regular meal patterns was associated with reduced malnutrition risk.” (Page 1, Lines 31-32).

Comment 2:
Introduction

  • Lines 37–44: Merge short and repetitive sentences regarding aging and malnutrition to create a more direct argument.
  • Lines 58–63: Remove '(figure 1)' from the middle of the sentence and reposition it more clearly at the end of the paragraph.
  • Line 62: Correct spacing ('northern Thailand ,') to 'Northern Thailand,'.

Response:

  • We have revised the Introduction for conciseness and improved grammar (Page 2, Lines 39–45).
  • The reference to “(Figure 1)” has been moved to the end of the paragraph for better readability (Page 2, Line 61).
  • The last paragraph of the Introduction has been revised for emphasize novelty of this research, and the comma had been removed (Page 2, Line 56-61).

Comment 3:

Materials and Methods

  • Lines 68–74: Simplify the description of geographic location by removing unnecessary travel details.
  • Lines 98–100: Correct repetition ('obtained using were collected using') to 'obtained using'.
  • Lines 117–122: Streamline sample size calculation text to remove repeated terms.
  • Lines 123–130: Add justification for statistical significance level or expected effect size.
  • Line 129: Replace 'P-value < 0.05' with 'p < 0.05' for consistency.
  • Line 133: Rephrase 'to allow for comparison of effect sizes' to 'to standardize variables for effect size comparison'.

Response: Thank you for your constructive comments.

  • The unnecessary travel details have been removed to simplify the description of the study location (Page 2, Line 67).
  • We have revised the sentence to correct repetition “Anthropometric measures (weight, height, BMI, mid-arm, waist, and calf circumference) were obtained using an electronic scale…” (Page 3, Lines 97-98).
  • The sample size calculation had been revised for more conciseness and reduce repetitive terms. (Page 4, Line 121-125)
  • The sample size was calculated using the infinite population proportion formula based on an estimated malnutrition prevalence of 11.25% among older adults in Northern Thailand, with a 95% confidence level and a 5% margin of error, which are standard parameters commonly used in cross-sectional epidemiological studies.
  • The term p < 0.05 has been used consistently throughout the manuscript instead of “P-value < 0.05.”
  • We have also revised the sentence “Macronutrient and micronutrient variables were standardized by dividing each by its respective standard deviations to allow for effect size comparison” (Page 4, Lines 135-137).

Comment 4:

Results

  • Lines 143–146: Standardize terminology, replacing 'well nutrition' with 'well-nourished'.
  • Lines 154–155: Simplify and break long sentences regarding eating episodes and fasting for better clarity.
  • Lines 158–164: Correct grammar: replace '. which' with ', which'.
  • Table 2 (Lines 183–189): Standardize p-value formatting and align columns properly.

Response: Thank you for your helpful comments.

  • The term “well nutrition” has been revised to “well-nourished” (Page 4, Line 147).
  • The sentence has been simplified for clarity to read: “Most participants consumed three meals per day, with a median overnight fasting duration of 13.5 hours.” (Page 4, Line 158-159).
  • The punctuation error has been corrected by replacing the period with a comma (Page 4, Line 163).
  • All tables have been standardized to use consistent p-value formatting, and column alignment has been corrected for clarity and readability (Page 5-8, Table 1-4).

Comment 5:

Discussion

  • Lines 206–211: Rewrite to remove unnecessary enumerations and improve flow.
  • Lines 240–245: Rephrase to be more concise ('Vitamin E is a lipid-soluble antioxidant, primarily α-tocopherol, that protects...').
  • Lines 258–265: Improve cohesion and separate the long sentence about sodium intake and adjustment effects into two.

Response:

  • We have shortened the sentences and reduced unnecessary enumerations to improve the flow of the discussion (Page 6, Lines 199–202).
  • The description of vitamin E has been rephrased for conciseness as: “Vitamin E is a lipid-soluble antioxidant, primarily α-tocopherol, that protects…” (Page 9, Line 244).
  • The section discussing excess sodium intake has been revised to improve cohesion and readability, and the long sentence has been separated for smoother flow (Page 9, Lines 273–276).

Comment 6:

Conclusion

Lines 295–304: Rephrase to make practical recommendations more explicit (e.g., 'promoting vitamin E-rich foods, avoiding prolonged fasting, and reducing sodium intake').

Response: We have revised the Conclusion to emphasize practical recommendations, including promoting vitamin E–rich foods, avoiding prolonged fasting, and reducing sodium intake (Page 10, Lines 323–329).

Comment 7:

References and Formatting
- References 6 and 45: Remove duplicates references
- References 8 and 364–366: Correct inconsistent capitalization.
- Ensure uniform formatting of journal titles, volumes, and page numbers.

Response: References were rechecked and duplicates removed. Format was revised to follow Nutrients guidelines.

Reviewer 2 Report

Comments and Suggestions for Authors

This study is commendable for its detailed assessment of patients’ dietary patterns and careful collection of data on physical activity and other relevant factors. The manuscript is clearly written, and the descriptions of the methodology are appropriate and well-organized. It is very interesting that SPPB scores were obtained for all participants.

However, the number of participants is far too small, falling below the minimum number of endpoints required for multivariate analysis—a major limitation. In addition, how were patients recruited? Were they consecutive patients seen in the hospital, or were participants selectively enrolled by physicians based on a presumed high likelihood of malnutrition?

Moreover, the content lacks novelty.

Author Response

Comment 1: Small sample size for multivariate analysis.
Response: We agree that the relatively small sample size is a limitation, resulting from a lower malnutrition prevalence than expected, which may have led to a low events-per-variable ratio in the multivariate model. This limitation has been explicitly acknowledged in the Discussion (Page 10, Lines 300–303).

Comment 2: Recruitment unclear.
Response:

  • We have clarified the recruitment process in the Methods Participants were recruited using convenience sampling from older adults attending primary care units and from community surveys, without selection based on specific characteristics or likelihood of malnutrition (Page 3, Lines 71–77).
  • In addition, the term “home visit” was replaced with “community survey” to indicate that the data collection was part of research activities rather than primary care services (Page 3, Line 71-77).

Comment 3: Novelty not sufficiently highlighted.
Response: We have added novelty statements in Introduction and Discussion, emphasizing that this is the first study in Thailand and one of few LMIC studies linking dietary intakes and meal patterns with malnutrition risk (Page 2, Lines 58–59; Page 10, Lines 313–315).

Reviewer 3 Report

Comments and Suggestions for Authors

The studies performed by the authors have revealed that among 200 community-dwelling older adults from northern Thailand, 23% were classified as at high risk of malnutrition (MNA < 24). Those at risk were older and more likely to be unemployed or have low income. Despite the absence of depressive symptoms, this group exhibited greater physical inactivity, cognitive impairment, and poorer physical performance. Mean daily energy intake in the group analysed was 1,609.8 ± 505.4 kcal, with only 38.5% achieving adequacy, whereas protein requirements were generally met. Intakes of several vitamins and minerals were insufficient, while sodium intake exceeded recommended limits in most participants. Higher intakes of carbohydrates, vitamin B6, magnesium, and vitamin E were inversely associated with malnutrition risk, though most associations were attenuated after adjustment for cognitive and physical function. Prolonged overnight fasting was consistently linked to an increased risk of malnutrition, whereas a higher number of daily eating episodes appeared to be protective. 
The manuscript is well written, the authors used an appropriate methodology, and combined results obtained from Survey questionnaires with nutrient aport, a fact which improve the quality of the interpretations. Results are presented concisely, and the conclusions sustain the results.
This study offers novel insights into malnutrition risk among community-dwelling older adults in Thailand by integrating detailed assessments of meal patterns, nutrient intake, and functional status. Unlike prior studies, this study revealed that overnight fasting duration and daily eating frequency are independently associated with malnutrition risk, underscoring the significance of eating behaviours beyond total energy intake. Adjusting for cognitive and physical function further elucidates how functional decline may mediate diet–nutritional status relationships. The persistent inverse association with vitamin E intake highlights a potential role for this micronutrient as a marker of malnutrition risk, suggesting new avenues for dietary interventions in this type of population.
Minor: 

1)A sentence cannot begin with an abbreviation. The authors must rewrite the sentence from R156.
2) The authors forgot to upload Supplement 1 to the MDPI platform.
3) The references are not written according to MDPI rules. 

Author Response

Comment 1: A sentence cannot begin with an abbreviation. The authors must rewrite the sentence from R156.
Response: We have carefully reviewed the entire manuscript and confirm that no sentence begins with an abbreviation.

Comment 2: The authors forgot to upload Supplement 1 to the MDPI platform.
Response: Thank you for pointing this out. The supplementary tables (Supplement 1) have been added and uploaded separately to the MDPI submission system.

Comment 3: The references are not written according to MDPI rules. 
Response: The reference list has been thoroughly revised and reformatted according to MDPI citation and style guidelines.

Reviewer 4 Report

Comments and Suggestions for Authors

This manuscript presents a well-designed and clearly written cross-sectional study assessing the relationship between dietary intake, meal patterns, and malnutrition risk among older adults in Northern Thailand. The study addresses an important and underexplored topic in geriatric nutrition, particularly within low- and middle-income country (LMIC) contexts. The authors employ validated instruments (MNA-FF, PHQ-9, Mini-Cog, SPPB) and appropriate statistical analyses to identify key dietary and behavioral factors associated with malnutrition. The results, particularly the associations between vitamin E intake, meal frequency, and overnight fasting, are novel and potentially impactful for public health strategies aimed at aging populations. The manuscript is well organized, adheres to MDPI format, and demonstrates methodological rigor. However, several issues require clarification, such as the following: 

  • Abstract: Include numerical results for key associations (e.g., vitamin E OR = 0.07, fasting OR = 1.94) to increase transparency and clarify that “meal patterns” refers specifically to overnight fasting duration, meal skipping, and eating frequency.

  • Introduction: Correct minor grammar: “... community-dwelling older adults remains limited, with most studies focusing on weight or metabolic outcomes.” and replace “figure1” with “Figure 1”.

  • Methods: Define the Thai baht-to-USD equivalent for readers unfamiliar with local currency and state whether BMI cutoffs followed WHO or Asian criteria.

  • Results:  Present p-values consistently to three decimal places and include 95% CIs in Table 4 for clarity on effect precision.

  • Discussion: Add one or two sentences connecting the findings to policy relevance (e.g., Thailand’s National Elderly Health Plan or community meal programs) and slight redundancy appears between lines 276–282; consider tightening.

  • References:  Ensure uniform journal abbreviations per Nutrients citation style (e.g., Nutr Rev instead of Nutrition Reviews).

Author Response

Comment 1:

Abstract: Include numerical results for key associations (e.g., vitamin E OR = 0.07, fasting OR = 1.94) to increase transparency and clarify that “meal patterns” refers specifically to overnight fasting duration, meal skipping, and eating frequency.
Response: We have included effect sizes (OR and 95% CI) for all key associations and clarified that “meal patterns” refer to overnight fasting duration, meal skipping, and eating frequency in the Abstract (Page 1, Lines 23–24, 27–31).

Comment 2:

Introduction: Correct minor grammar: “... community-dwelling older adults remains limited, with most studies focusing on weight or metabolic outcomes.” and replace “figure1” with “Figure 1”.
Response: Thank you for your suggestion. We have revised the paragraph to correct grammar and improve flow, and have corrected “figure1” to “Figure 1” (Page 2, Lines 56–61).

Comment 3:

Methods: Define the Thai baht-to-USD equivalent for readers unfamiliar with local currency and state whether BMI cutoffs followed WHO or Asian criteria.
Response:

  • The currency conversion has been added in Table 1 (5000 baht/month = 153 USD/month) (Page 5, Table 2).
  • BMI cutoffs were not separately defined, as anthropometric data were used for MNA score calculation. However, studies in Thailand typically adhere to Asian BMI criteria. (Page 3, Line 99).

Comment 4:

Results: Present p-values consistently to three decimal places and include 95% CIs in Table 4 for clarity on effect precision.
Response: All p-values have been standardized to three decimal places and applied consistently throughout the manuscript. In addition, 95% confidence intervals have been added to Table 4 to improve transparency of effect estimates (Pages 5–8, Tables 1–4).

Comment 5:

Discussion: Add one or two sentences connecting the findings to policy relevance (e.g., Thailand’s National Elderly Health Plan or community meal programs) and slight redundancy appears between lines 276–282; consider tightening.
Response: Thank you for your suggestion. We have revised the paragraph to make the policy relevance more explicit and improved the paragraph’s clarity and flow by removing redundancy (Pages 10, Lines 289–296).

Comment 6:

References: Ensure uniform journal abbreviations per Nutrients citation style (e.g., Nutr Rev instead of Nutrition Reviews).

Response: The reference list has been thoroughly revised and reformatted according to MDPI and Nutrients citation style guidelines, ensuring consistent use of journal abbreviations.

Reviewer 5 Report

Comments and Suggestions for Authors

This is an interesting study based on an important topic of malnutrition and frailty in older populations, however some concerns need to be addressed to ensure the results accurately reflect the conclusions. 

The conceptual framework (figure 1) for determinants of malnutrition in older adults appears somewhat incomplete. The authors mention dietary behaviours but what about other health-related behaviours such as smoking or alcohol intake? Also under the umbrella term physiological factors the authors list "aging process" but this is too vague and would benefit from more detailed information such as decreased sense of taste/smell or poor oral health, reduced mobility etc.  Overall, this conceptual framework needs to be adapted to reflect current well documented determinants of malnutrition.

Eating behaviour assessments: given what we know about the aging process and memory, do the author believe a 24 hour recall is the most suited method to capture all foods and snacks and beverages consumed? Please elaborate on this, and at the very least,  refer to it as a significant limitation in the discussion.

It is not quite clear when the FFQ was used – was this in all participants or just in those who reported typically eating more or less than what was reported in the 24HR? I would have expected this to have (ideally) been used in all participants.

The authors are praised for carrying out a sample size calculation and using many validated data collection tools.

The authors have looked at unemployment and in Table 1 illustrate a significant difference between high risk of malnutrition and being well nourished. With no personal awareness of employment rates in such an age group in Thailand, the question needs to be asked whether this is relevant in what is generally considered a retired proportion of the population.

Do the authors have a plausible explanation why vitamin E was higher in the well nourished group versus the group with high malnutrition risk? Can the dietary and eating behaviour data shed light on this? This is important to address in the discussion as it is one of the main findings.

The ethical approval information should also be added to the methodology section.

Comments on the Quality of English Language

The English language needs to be improved in parts throughout the manuscript. 

Author Response

Comment 1: The conceptual framework (figure 1) for determinants of malnutrition in older adults appears somewhat incomplete. The authors mention dietary behaviours but what about other health-related behaviours such as smoking or alcohol intake? Also under the umbrella term physiological factors the authors list "aging process" but this is too vague and would benefit from more detailed information such as decreased sense of taste/smell or poor oral health, reduced mobility etc.  Overall, this conceptual framework needs to be adapted to reflect current well documented determinants of malnutrition.
Response: Thank you for your valuable suggestion. Figure 1 has been revised to provide a more comprehensive conceptual framework. We have added details regarding the aging process (e.g., decreased sensory function, anorexia), disease-related malnutrition, and included additional health-related behaviors such as smoking and alcohol consumption. The revised figure also emphasizes the three key mechanisms contributing to malnutrition (Page 2, Figure 1).

Comment 2: Eating behaviour assessments: given what we know about the aging process and memory, do the author believe a 24 hour recall is the most suited method to capture all foods and snacks and beverages consumed? Please elaborate on this, and at the very least, refer to it as a significant limitation in the discussion.
Response: We agree with this important point. Since 24-hour recall may be less reliable among older adults due to memory limitations, we used probing techniques and picture aids to enhance recall accuracy during data collection. We have also added a note in the Limitations section to acknowledge potential recall bias associated with this method (Page 10, Lines 298–303).

Comment 3: It is not quite clear when the FFQ was used – was this in all participants or just in those who reported typically eating more or less than what was reported in the 24HR? I would have expected this to have (ideally) been used in all participants.
Response: We apologize for the unclear description. The Food Frequency Questionnaire (FFQ) was administered to all participants alongside the 24-hour dietary recall. When participants reported unusual or infrequently consumed foods, research staff verified and adjusted the reported frequency of those specific items to improve accuracy (Page 3, Lines 104-111).

Comment 4: The authors have looked at unemployment and in Table 1 illustrate a significant difference between high risk of malnutrition and being well nourished. With no personal awareness of employment rates in such an age group in Thailand, the question needs to be asked whether this is relevant in what is generally considered a retired proportion of the population.
Response: Thank you for this thoughtful comment. In Thailand, approximately 30–40% of older adults remain economically active beyond retirement age, particularly in informal worker such as agricultural work. Employment status is therefore still a relevant socioeconomic indicator in this population. However, because unemployment was strongly correlated with other variables such as income, education level, and health insurance, we did not include it in the multivariable analysis to avoid overadjustment.

Comment 5: Do the authors have a plausible explanation why vitamin E was higher in the well nourished group versus the group with high malnutrition risk? Can the dietary and eating behaviour data shed light on this? This is important to address in the discussion as it is one of the main findings.
Response: Thank you for pointing this out. Interestingly, vitamin E intake was significantly higher among well-nourished participants compared with those at high risk of malnutrition, despite similar total fat intake. This likely reflects differences in the sources and quality of dietary fat. Older adults with better nutritional status may consume more plant-based oils, nuts, and seeds, which are rich in vitamin E, whereas those at higher risk may rely more on animal fats or fried foods with lower vitamin E content. This explanation has been incorporated into the Discussion section (Page 9, Line 260-267).

Comment 6: The ethical approval information should also be added to the methodology section.
Response: The ethics approval statement has been added to the Methods section (Page 3, Lines 78–80).

Comment 7: The English language needs to be improved in parts throughout the manuscript. 
Response: The entire manuscript has been professionally edited to improve English language, grammar, and overall readability.

Round 2

Reviewer 1 Report

Comments and Suggestions for Authors

After review, the authors responded to all questions, comments, and revisions, so I consider the article acceptable in its current form.

Reviewer 5 Report

Comments and Suggestions for Authors

The authors have made the required changes

Comments on the Quality of English Language

The English language needs to be improved in parts throughout the manuscript.